# A Ruptured Tri-Lobulated ICA–PCom Aneurysm Presenting with Preserved Neurological Function: Case Report and Clinical–Anatomical Analysis

**DOI:** 10.3390/diagnostics16010073

**Published:** 2025-12-25

**Authors:** Stefan Oprea, Cosmin Pantu, Alexandru Breazu, Octavian Munteanu, Adrian Vasile Dumitru, Mugurel Petrinel Radoi, Daniel Costea, Andra Ioana Baloiu

**Affiliations:** 1Faculty of General Medicine, “Carol Davila” University of Medicine and Pharmacy, 050474 Bucharest, Romania; 2Department of Anatomy, “Carol Davila” University of Medicine and Pharmacy, 050474 Bucharest, Romania; 3Department of Pathology, Faculty of Medicine, “Carol Davila” University of Medicine and Pharmacy, 030167 Bucharest, Romania; 4Department of Vascular Neurosurgery, National Institute of Neurology and Neurovascular Diseases, 077160 Bucharest, Romania; 5Department of Neurosurgery, “Victor Babes” University of Medicine and Pharmacy, 300041 Timisoara, Romania; 6Doctoral School, “Carol Davila” University of Medicine and Pharmacy, 050474 Bucharest, Romania; 7Puls Med Association, 051885 Bucharest, Romania

**Keywords:** ICA–PCom aneurysm, subarachnoid hemorrhage, multilobulated aneurysm morphology, microsurgical clipping, cerebrovascular anatomy, aneurysm rupture dynamics, cisternal hemorrhage, neurosurgical aneurysm management

## Abstract

**Background and Clinical Significance:** Although rupture of aneurysms at the internal carotid-posterior communicating artery (ICA-PCom) junction accounts for a small percentage of all ruptured intracranial aneurysms, they are clinically relevant due to their proximity to perforator-rich cisterns, the optic-carotid-oculomotor pathways and flow-diverting zones, as well as their high likelihood for causing early neurological instability. Additionally, ruptured ICA-PCom aneurysms that have multiple lobulations are associated with increased variability in wall shear stress, local inflammatory remodeling and higher propensity for rupture at smaller sizes compared to other types of aneurysms. Due to the rapidity of early physiological destabilization in most patients with ruptured ICA-PCom aneurysms, clinical–anatomical correlations in these cases are often obscured by neurological deterioration; therefore, the presentation of this patient provides a unique opportunity to correlate the minimal early symptoms, tri-lobulation of the aneurysm and confined cisternal hemorrhage, to better understand rupture behavior, surgical decision-making in an anatomically challenging area, and postoperative recovery. **Case Presentation:** A 48-year-old hypertensive female experienced an acute “thunderclap” headache accompanied by intense photophobia and focal meningeal irritation, but, unexpectedly, retained a normal neurologic examination. She did exhibit some minor ocular motor micro-latencies, early cortical attentional strain and lateralized pain sensation that suggested localized cisternal involvement despite lack of generalized neurologic impairment. Digital subtraction angiography and three-dimensional CT angiography revealed a ruptured, tri-lobulated aneurysm originating from the communicating portion of the left internal carotid artery proximal to its origin from the posterior communicating artery, oriented toward the perimesencephalic cisterns. The aneurysm was surgically clipped using a standard left pterional craniotomy with direct visualization, after careful dissection through the carotid and optic windows to preserve the anterior choroidal artery, oculomotor nerve, and surrounding perforators. The neck of the aneurysm was reconstructed with a single straight clip, without compromise to the parent vessel lumen. The patient had an uneventful postoperative course without vasospasm or neurologic deficit. At both 3 and 9 months postoperatively the patient remained free of clinical neurologic deficit, and imaging demonstrated continued aneurysm exclusion, preserved ICA-PCom anatomy, and no evidence of delayed ischemic injury or hydrocephalus. **Conclusions:** The goal of this report is to demonstrate how a ruptured, morphologically complex ICA-PCom aneurysm may present with preserved neurologic function, thereby enabling the study of clinical–anatomical associations before secondary injury mechanisms intervene. The relationship between the configuration of the patient’s symptoms, geometry of the aneurysm and pattern of hemorrhage within the cisterns offers insight into a rare rupture pattern observed during routine clinical experience. Through complete anatomical analysis, timely microsurgical reconstruction and consistent follow-up, the authors were able to achieve long-term recovery of this particular patient. Continued advancements in vascular imaging techniques, aneurysmal wall modeling, and postoperative monitoring will likely help clarify the underlying mechanism(s) responsible for such presentations.

## 1. Introduction

Aneurysmal subarachnoid hemorrhage is a less frequent cerebrovascular emergency; however, it has a greater impact on acute neurologic decline than other types of emergencies. Many times, initial evaluations are limited because of an immediate increase in intracranial pressure, diffuse cisternal hemorrhage, fluctuating levels of consciousness, and secondary physiologic effects that mask the primary effect of an arterial rupture. Therefore, opportunities to assess clinical–anatomical correlations soon after the rupture occur very seldomly, especially if the aneurysm is morphologically similar to one with a high rupture propensity [1].

Internal carotid-posterior communicating artery (ICA-PCom) junction aneurysms represent a particularly delicate subset of aneurysms. While representing a small percentage of all intracranial aneurysms, they are represented disproportionately higher in ruptured lesions and have been associated with oculomotor palsy, irritation of the optic pathways, and deficits resulting from perforator injury [2]. Due to their location within very narrow cisternal spaces, these aneurysms are positioned near areas that respond dramatically to minimal amounts of bleeding: the opticocarotid recess, the oculomotor nerve tract, the anterior choroidal artery, and densely grouped hypothalamic and mesencephalic perforating vessels. When rupture occurs in this area, marked symptomatology and early neurologic instability usually result. The difficulty in achieving such assessments is further compounded by an evaluation of the aneurysm’s morphology [3]. It is well known that multilobulated, irregular aneurysms specifically those with bi- or tri-lobulated domes are at increased risk of rupture due to unstable shear forces, heterogeneous wall remodeling, and focal areas of structural degradation. These aneurysms rupture at much lower diameters, tend to distribute blood throughout multiple complex cisternal vectors, and are almost exclusively associated with either severe or rapidly worsening clinical presentations. The presence of a multilobulated ICA-PCom aneurysm with a significant irregularity and no evidence of deficit upon neurological examination after rupture is therefore extremely rare, since the morphological risk factor typically correlates with more extensive or rapidly progressive deficits [4]. The likelihood of documenting such a case is significantly reduced when evaluating both clinical and anatomical aspects. In most instances, patients with ruptured ICA-PCom aneurysms will have deteriorated prior to the ability to establish correlations between clinical and anatomical findings, and secondary injury will frequently obscure the subtle characteristics indicative of the rupture direction or the cisternal distribution of blood dissemination [5]. Recording a case in which meningeal traction pain, photophobia, and delicate cranial nerve micro-latencies can be correlated with a tri-lobulated, posteriorly directed aneurysm at the ICA-PCom junction is an opportunity that is generally not available to clinicians practicing in a regular setting.

As such, this case offers an uncommon opportunity to evaluate a “window” of time in which a clinically high-risk aneurysm ruptured in a confined cisternal space while the patient was neurologically intact and able to allow for detailed clinical–anatomical correlations to be made before secondary mechanisms become involved. The subsequent uneventful course of events for the patient—including stable vascular patency and complete clinical recovery—distinguishes this case within a population in which delayed ischemic deficits, vasospasm, and hydrocephalus are commonly observed as sequelae.

The purpose of this manuscript is to provide a clear, objective description of this unusual combination: to detail the patient’s clinical presentation, to relate the subtle differences in the patient’s neurological findings to the aneurysm’s complex morphology, to detail the surgical reconstruction performed with consideration to the anatomical relationship of the various cisternal structures, and to provide documentation of the patient’s postoperative course in a manner that may assist clinicians who encounter similar complex configurations of aneurysms.

## 2. Case Presentation

A 48-year-old female patient with a history of hypertension presented to the Emergency Department with an abrupt onset of “thunderclap” headache, which she described as being holocranial and reached maximum intensity (10/10 on the Numeric Rating Scale) in approximately one second. She also reported persistent nausea, forceful vomiting, increased sensitivity to sensory stimuli, severe photophobia, even in very low ambient lighting, a diffuse sensation of pressure in her head, and her headache began immediately. The patient’s presentation was sudden, had maximal intensity at onset and included vegetative symptoms which met the high probability clinical criteria for subarachnoid hemorrhage (SAH) based on the Ottawa Subarachnoid Hemorrhage Rule. At time of evaluation, the patient’s GCS was 15 (E4 V5 M6). Although the patient was conscious and verbally responsive, the patient’s clinical picture was deemed high-risk for aneurysmal SAH, necessitating immediate vascular imaging and consultation with Neurosurgery.

Based upon previously established SAH grading systems, the initial severity of the patient’s SAH was determined to be a Hunt-Hess Grade 1 and WFNS Grade 1. The NIHSS score of the patient was 0. A detailed neurological evaluation was conducted, including assessment of the patient’s mental status, cranial nerves, motor function, sensory function, coordination, and meningeal signs. Non-validated composite scores were not utilized in this evaluation.

The patient was alert, cooperative and fully oriented, able to speak fluently, and had intact comprehension. However, the extreme nature of the patient’s pain and photophobia limited the bedside cognitive endurance testing, which was conducted; the patient responded appropriately to command, did not demonstrate clinical evidence of aphasia, neglect or obvious executive dysfunction. Slowness in beginning task initiation and decreased ability to sustain attention were noted primarily as symptomatic performance limits. Despite these limitations, brief standardized screening assessments were administered to the patient; the patient’s Digit Span Backward scaled score was 8 (low-average range), indicating decreased working memory performance in the acute phase. Trail Making Test part A was completed by the patient in 48 s (slowed relative to age and education level) and the patient reported increasing headache pain while performing visually guided tracking, which provided support for a practical interpretation that the symptom burden significantly limited the patient’s ability to sustain attention under bedside conditions.

Clinical meningeal signs were evident in the patient. The patient exhibited resistance and significant pain radiating to the vertex and left frontotemporal region with passive neck flexion; even minimal flexion caused the patient to exhibit observable discomfort with nociceptive response (facial tightening, brief breath-holding, truncal co-contracture). Due to the presence of pain, Kernig- and Brudzinski-type maneuvers could not be completely elicited.

Photophobia was documented both clinically and through the use of validated instruments. The Photophobia Severity Index (PSI) was 17/20 (severe), and the Visual Light Sensitivity Questionnaire-8 (VLSQ-8) was 32/40 (indicating severe and functionally limiting light sensitivity) in the acute phase. The cranial nerve examination was grossly intact: pupils were equal and reacted to light; extraocular movements were complete; facial symmetry was preserved; and bulbar function was normal. Due to the inability to conduct prolonged testing, minor delays and brief interruptions in convergence-accommodation were documented descriptively and were not interpreted as localized signs. Serial King-Devick testing demonstrated a decrease in the patient’s rapid number-naming performance from 45 s at baseline (no errors) to 68 s on repeated testing (4 errors), indicating worsening visual-cognitive efficiency under fatigue and symptom burden, and providing support for the bedside impression of decreased tolerance for sustained oculomotor/visual tasks without indicating focal structural dysfunction.

The motor examination demonstrated 5/5 strength in all extremities without drift; sensation to light touch was symmetric. Coordination testing (finger-nose-finger; heel-knee-shin) demonstrated no dysmetria or clear cerebellar dysfunction. Rapid alternating movements were symmetric and mildly slowed overall, consistent with pain and fatigue. The patient was unable to ambulate due to the severity of her symptoms; however, the patient maintained axial control and seated posture. Fundoscopy demonstrated Stage II hypertensive retinopathy, with arteriolar narrowing and altered arteriovenous ratios indicative of chronic hypertensive vasculopathy. Collectively, the instantaneous onset of the patient’s headache, the pronounced meningeal signs, and the severe photophobia supported the need for urgent vascular imaging to identify an aneurysmal source and provide timely definitive treatment to minimize the risks of rebleeding, vasospasm, and delayed ischemic injury.

Pattern recognition was utilized to organize the potential differential diagnoses.

•Most likely diagnosis: Aneurysmal subarachnoid hemorrhage (SAH); based on the characteristics of the headache, the intensity of pain at onset, the pronounced meningeal signs and photophobia, and the lack of focal neurological deficits (Hunt–Hess 1; WFNS 1; NIHSS 0).•Second most likely diagnosis: Non-aneurysmal perimesencephalic SAH; this is less likely given the total clinical intensity of the presentation and the nature of the pain-provoking mechanisms.•Third most unlikely diagnosis: Reversible cerebral vasoconstriction syndrome (RCVS); unlikely based on the absence of multiple episodes of thunderclap headaches and typical precipitating events;•Fourth least likely diagnosis: Thrombosis of the cerebral venous sinuses; unlikely given the absence of papilledema, seizures, and progressive clinical deterioration;•Fifth least likely diagnosis: Dissection of a cervical artery; unlikely given the absence of focal pain syndrome or cranial nerve deficits;•Sixth least likely diagnosis: Infectious meningitis; unlikely given the instantaneous onset of symptoms, afebrile state, and the absence of preceding symptoms.

Given the necessity for structural resolution to support the clinical impressions made from her neurological examination, preoperative catheter angiography was performed. Selective injections of both internal carotid arteries and the left vertebral artery identified an anatomically solitary but morphologically complex aneurysm located at the communicating segment of the left internal carotid artery at the posterior communicating artery origin (Figure 1A,B). The aneurysm was triangular in shape with a neck approximately 3 mm in length and a maximal sac diameter of approximately 10 mm, projecting posteriorly and slightly laterally into the carotid and interpeduncular cisterns. The irregular, multi-lobulated shape of the aneurysm was consistent with localized wall remodeling and rupture-prone hemodynamic stress, features that were consistent with the sudden, maximal-at-inception headache she described at the time of her initial presentation.

The preoperative 3D rotational angiography/CTA revealed that the ICA-PCom aneurysm was located in the posterior and slightly medial aspect of the ICA, had a tri-lobed sac (1 larger posterior lobe and 2 smaller satellite lobes), and an aneurysm dome height of approximately 10 mm. The aneurysmal neck diameter was ~3 mm based on the 3D reconstructed images of the aneurysm (Figure 2A,B; Figure 3A,B). The AChA was found to originate superior to and distal to the aneurysm neck plane, and the PCom artery also originated from the aneurysm base but could be identified proximal to its origin. These findings suggested that anatomical preservation of these arteries could be achieved during the construction of the neck. The parent ICA demonstrated no evidence of calcification, thrombosis or wall irregularities, and the neck was well defined, indicating a low likelihood of clip slippage and a predictable course for dissection.

Following a craniotomy, a left pterional craniotomy was performed under general anesthesia to create an open path for a low basal trajectory toward the supraclinoid ICA and ICA-PCom junction. The area was exposed by performing a frontotemporal craniotomy and drilling the sphenoid ridge. Cisternal CSF was released via a proximal Sylvian fissure to relax the brain and assist in exposing the opticocarotid and carotid ocular window areas. The supraclinoid ICA was circumferentially dissected and skeletonized from the distal dural ring to the bifurcation with preservation of posterior wall perforators and proximal PCom, providing safe proximal control and establishing the neck plane.

During surgery, it was confirmed that the aneurysm was a tri-lobulated, posteriorly directed sac, approximately 10 mm in length, with a 5 mm neck located on the posterior wall of the communicating ICA segment at the origin of the PCom. The key technical challenge associated with this case was the branch relationship of the aneurysm. Specifically, the PCom originated directly from the neck of the aneurysm, whereas the AChA originated from a separate ostium above the neck and traveled postero-inferiorly along the optic tract in close proximity to the superior edge of the neck. Following arachnoid dissection along natural planes, selective clot removal to expose the posterior aneurysm surface without traction, the PCom was identified and protected by being followed posteriorly to identify perforators, and the AChA was visually inspected to determine if there was adherence or incorporation into a lobe of the aneurysm.

A temporary clip was placed on the ICA to reduce turgor and improve neck flexibility. Using a 5.2 mm straight Yasargil clip, definitive exclusion of the aneurysm was obtained by placing the clip parallel to the ICA, advancing the clip to completely enclose the neck of the aneurysm, and preserving patency of the ICA, PCom, and AChA. After removing the temporary clip, direct microscopic observation of the vessels demonstrated preserved vessel diameter and pulsatility, complete collapse of the aneurysm dome, no remaining neck (“dog ear”), and intact visible perforators. The dura mater was then closed tightly, and the wound was closed in multiple layers with an epidural drain.

Post-operatively, the patient was awake and neurologically intact (alert, oriented; pupils equal and responsive; extraocular movement intact; all four limbs strong and no drift; no sensory, speech, or visual field deficits). All serial exams throughout the patient’s early recovery were stable, and there were no clinical concerns regarding early vasospasm or decline. Due to the stable neurological exam and uncomplicated cisternal operative course (no parenchymal dissection; no intraoperative bleeding; no vascular damage; and progressive brain relaxation after cisternal decompression), immediate post-operative cranial CT was delayed until follow-up or if the patient demonstrated clinical decline. The patient’s hospitalization was uneventful: no worsening of photophobia or headache; no confusion; no localized deficits suggesting delayed cerebral ischemia; and no laboratory abnormalities indicative of early hyponatremia or Syndrome of Inappropriate Antidiuretic Hormone Secretion (SIADH). The patient mobilized gradually and returned home on post-operative day 7 with instructions on managing blood pressure, maintaining adequate hydration, and returning to graded levels of physical activity.

By three months post-operatively, the patient had completed resolution of their pre-operative symptoms, and their neurological exam was normal. A cranial CT revealed the aneurysm clipping to be in a stable position; basal cisterns to have expanded; and no evidence of infarction or hydrocephalus (Figure 4 A,C). By nine months post-operatively, the patient continued to be neurologically intact and experienced no headaches, photophobia, disequilibrium, cognitive dysfunction, or late cranial nerve deficits. Another cranial CT demonstrated the aneurysm clipping to remain in a stable position; normal-sized ventricles; fully expanded cisterns; and reabsorption of previous subarachnoid hemorrhage (SAH) products, with no radiographic evidence of delayed ischemia or hydrocephalus (Figure 5 A,B); demonstrating long-term durability of aneurysm exclusion and preservation of the parent vessel.

The case report represents one example of how rupture of an ICA-PCom aneurysm can occur with the presence of significant meningeal symptoms but relatively minor focal neurologic deficits, thus potentially masking the severity of the SAH. Although the anatomic location of the posteriorly directed aneurysm explains the distribution of the basal cisternal hemorrhage (and possibly contributed to her photophobia), along with the meningeal traction and pain she experienced, her lack of frank cranial nerve palsies is also consistent with a lower clinical grade presentation.

It is important to recognize that this case report should not suggest a predictable pattern; it clearly demonstrates that grading based upon deficit may provide a false sense of security regarding urgency for vascular imaging and definitive securement, even if the meningeal features and time-course indicate otherwise.

When considering the treatment options for ruptured ICA-PCom aneurysms, treatment options should still remain individualized. With regard to the patient described in this case, the tri-lobulated aneurysm morphology, its neck configuration and proximity to the origin of the PCom artery, along with the AChA artery originating from a position immediately above the neck favored microsurgical clipping in this case. Microsurgical clipping allows for the ability to directly visualize the aneurysm and reconstruct the vessel while protecting adjacent arterial branches and perforating vessels. There are many instances in which endovascular therapy will be the preferred method in certain anatomical or institutional settings. Thus, this case report should be viewed as an anatomically driven rationale in a single case versus a recommendation for a specific therapy modality.

As with all case reports there are inherent limitations associated with such a report. It is likely that the patient’s favorable outcome was due to a combination of both patient and lesion-specific characteristics that are not applicable to all patients. However, the fact that follow-up imaging demonstrated no evidence of recurrence and the patient achieved full resolution of her symptoms supports the durability of definitive exclusion in this case. Conversely, the initial presentation of this patient again highlights the importance of prioritizing timely evaluation of patients with thunderclap headache symptoms and meningeal features over evaluation of focal neurological deficits.

## 3. Discussion

Although a rupture of an ICA-PCom aneurysm is a relatively common source of aSAH with potential for significant symptoms due to hemorrhage in compact basal cisterns, significant symptoms can occur with little blood evident radiographically [6]. In addition, the described case presents one possible clinical manifestation of a ruptured ICA-PCom aneurysm: abrupt thunderclap headache with pronounced photophobia and meningeal signs without focal neurological findings and with low initial clinical grade. Although a “low-grade” aSAH presentation with a lack of focal neurologic findings can be misleading when evaluated with deficit-based grading systems, the headache’s onset and meningeal findings necessitated prompt vascular imaging and rapid aneurysm securing to minimize the risk of early rebleeding [7]. Additionally, the described case should be viewed as an example of a specific phenotype and not as indicative of a recurring pattern or predictable clinical course. Table 1 intends to summarize key, practice-relevant evidence directly supporting the clinical and management points illustrated by this case.

One practical application of this case is that a bedside phenotype can be documented early—prior to complications (hydrocephalus, sedation, vasospasm, etc.) or treatment-related factors (evolving ischemic injury), which potentially limit reliable examination [19]. In this case, the predominant symptom complex (severe photophobia and meningeal signs with preserved focal function) was a clinically relevant indicator that prompted rapid escalation to definitive imaging and treatment. Documenting the bedside phenotype prior to complications may provide additional clinicoradiologic correlation for low-grade presentations, however subsequent clinical decline is a frequent occurrence and may obscure the initial neurological findings.

In addition, the decision regarding whether to pursue microsurgical versus endovascular treatment for ruptured ICA-PCom aneurysms should continue to be made individually, taking into account the aneurysm’s geometric characteristics (size, lobulation, neck configuration), relationship to branches and perforators, rupture status, patient-specific considerations, and institution-specific expertise. For the described case, the tri-lobulated aneurysm configuration, broad neck characteristics, and close proximity to perforator-rich areas provided strong support for microsurgical clipping of the aneurysm, where direct visualization of the neck, parent vessel, and perforators allows for definitive exclusion of the aneurysm from circulation while preserving critical structures [20]. However, there are other anatomical and institutional situations, where endovascular approaches would be equally suitable for treating ruptured ICA-PCom aneurysms, particularly when favorable neck morphology exists and durable occlusion can be obtained using currently available techniques and expertise [21,22]. Therefore, this report does not advocate for either modality but rather emphasizes the importance of anatomy-driven treatment decisions in the acute setting.

Although the described patient had a successful outcome, the overall risk associated with a rupture is still affected by the location of hemorrhage, aneurysm morphology, and local anatomical relationships, and includes risks of rebleeding, vasospasm/delayed cerebral ischemia, hydrocephalus, and cranial nerve dysfunction [23]. The fact that no complications were observed in the described case does not represent the majority of patients with ruptured ICA-PCom aneurysms, but rather demonstrates that high symptom severity can co-exist with preserved focal examination and low clinical grade, thereby emphasizing the necessity for timely diagnosis and securement of the aneurysm regardless of the initial clinical score [24].

Additionally, this report illustrates a fundamental shortcoming of morphological risk assessment frameworks, which generally utilize descriptive terms (e.g., lobulation, size, direction of projection) to quantify rupture risk and symptom expression, yet cannot adequately capture the nuances of rupture risk or symptom expression in low-grade presentations with preserved focal function [25]. As a consequence, AI-driven and computational approaches that include high-resolution vascular geometry, quantitative shape descriptors, and data-driven pattern recognition are being investigated to improve risk characterization and decision-making, though challenges to validation, interpretation, and implementation exist [26]. Incorporating anatomically detailed and well-phenotyped examples of low-grade but symptomatically active rupture, similar to the case presented here, may ultimately strengthen future predictive models for clinically important phenotypes that are frequently underrepresented in extant datasets [27].

As with all case reports, the inherent limitations of a single-patient report apply. The clinical course and successful outcome likely depend upon a variety of patient- and lesion-specific factors that do not generalize to others. Nonetheless, the described case has practical utility in documenting a severe symptom phenotype with preserved focal neurological function, supports rapid imaging in cases of suspected high-risk thunderclap presentations, and provides a rationale for selecting a microsurgical approach when aneurysm geometry and branch/perforator relationships suggest that endovascular exclusion of the aneurysm will be less certain.

## 4. Conclusions

The opportunity to study a structurally complicated aneurysm at the ICA-PCom junction presented an interesting example of how a large, complicated aneurysm with many lobes could present itself and progress clinically after a rupture occurred, when significant portions of the patient’s initial neurological status were intact. Therefore, several relationships between the aneurysm’s configuration and the location of the bleed were evident and more readily discernible than would be expected in the usual rupture pattern of simpler aneurysms. Her relatively stable post-operative course and stable long-term imaging reflect her own course and cannot be used to generalize to other similar cases; however, the details of her anatomy and course may provide useful information to other clinicians caring for similar complicated aneurysms.

In the future, further advances in high-resolution imaging techniques, further improvements in models of biomechanics of aneurysmal walls, and the increasing interest in molecular markers of wall instability are likely to lead to improved understanding of the biological mechanisms involved in rupture in anatomically challenging areas (such as the ICA-PCom junction). Advances in intra-operative monitoring of patients during surgical and interventional procedures, the use of pharmacologic agents to modulate the initial inflammatory response to injury, and additional experience combining microsurgical and endovascular techniques will continue to influence the management of patients with complex aneurysms such as those seen at the ICA-PCom junction.

## Figures and Tables

**Figure 1 diagnostics-16-00073-f001:**
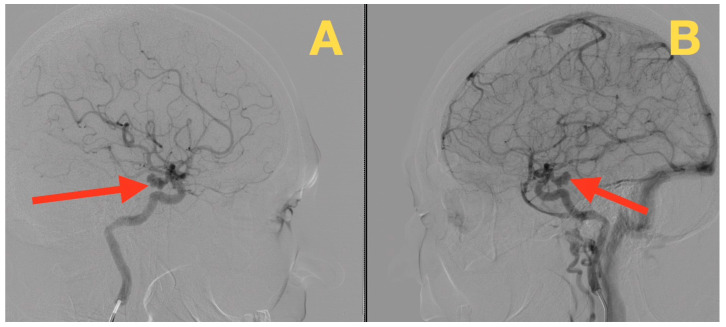
Preoperative digital subtraction angiography. (**A**) Anteroposterior projection of the left ICA demonstrates a tri-lobulated saccular aneurysm arising from the communicating (C7) segment at the ICA–PCom junction (red arrow). (**B**) Lateral projection confirms a posteriorly projecting dome at the ICA–PCom junction (red arrow). No additional aneurysms or flow-limiting stenoses are identified.

**Figure 2 diagnostics-16-00073-f002:**
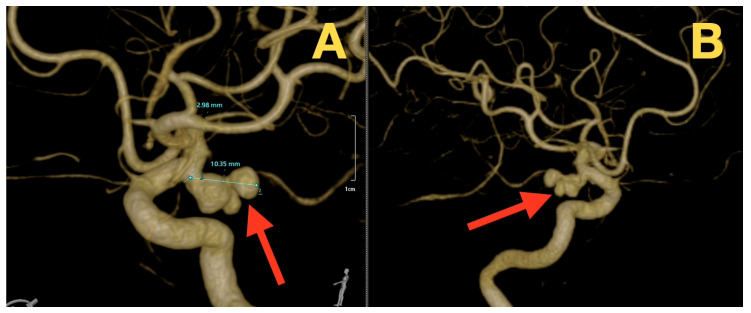
Three-dimensional rotational CT angiography. (**A**) Volume-rendered 3D reconstruction demonstrates a tri-lobulated ICA–PCom aneurysm with a neck measuring ~3 mm (red arrow) and posterior/slightly medial projection. (**B**) Oblique projection shows the maximal dome dimension (~10 mm) and the relationship between the aneurysm neck and the anterior choroidal artery (AChA) (red arrow), defining the operative corridor.

**Figure 3 diagnostics-16-00073-f003:**
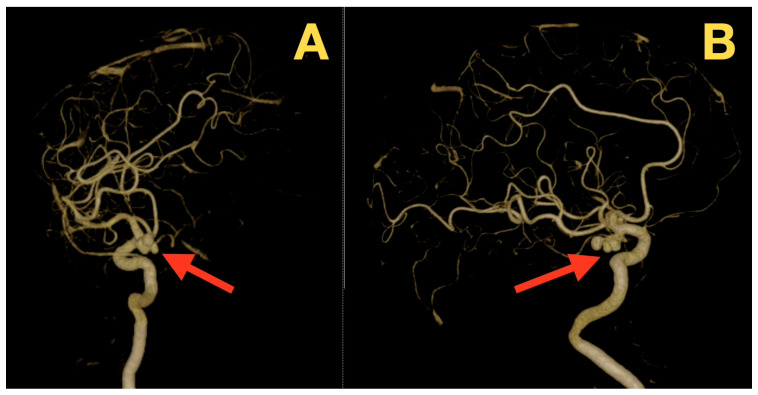
Additional rotational projections. (**A**) Superior–posterior projection highlights the posterior dome and two smaller lobules (red arrow), confirming multilobulation. (**B**) Lateral–oblique projection illustrates the aneurysm–parent ICA curvature and PCom origin relationship (red arrow), relevant for clip-vector planning while preserving branch patency.

**Figure 4 diagnostics-16-00073-f004:**
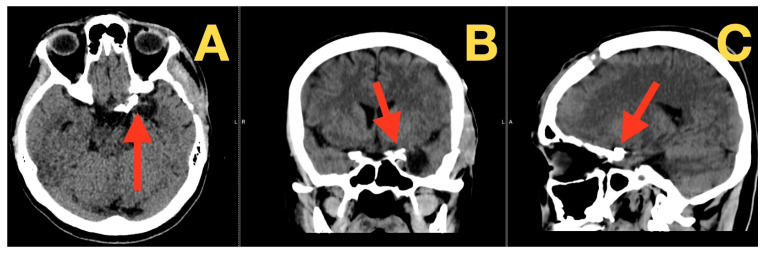
Three-month postoperative CT scan. (**A**) Axial CT demonstrates stable clip position along the posterior wall of the left supraclinoid ICA (red arrow), with no infarct, hematoma, or hydrocephalus. (**B**) Coronal reconstruction shows re-expansion of basal cisterns and preserved parenchymal architecture (red arrow), without mass effect or delayed ischemic change. (**C**) Sagittal view confirms clip positioning at the ICA–PCom junction (red arrow) with normal ventricular size and no complications.

**Figure 5 diagnostics-16-00073-f005:**
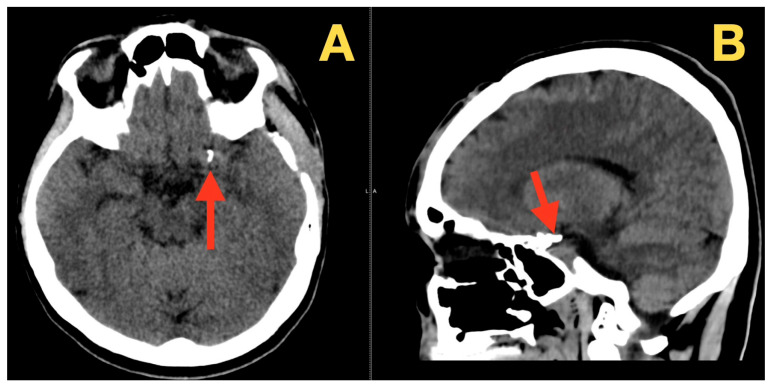
Nine-month postoperative CT scan. (**A**) An axial CT image showing the continued stability of the clip position along the posterior wall of the left supraclinoid ICA (red arrow), with maintained basal cisternal patency and no evidence of late infarction, hematoma, or hydrocephalus. (**B**) A sagittal reconstruction showing the maintenance of the skull base and parasellar anatomy (red arrow), normal ventricular size, and no new vascular-appearing lesion adjacent to the clip construct. In combination with the patient’s complete clinical recovery, these findings indicate a durable exclusion of the aneurysm and persistent patency of the parent artery on long-term follow-up.

**Table 1 diagnostics-16-00073-t001:** Targeted literature context directly supporting the clinical and management points illustrated by this case (ruptured ICA–PCom aneurysm with high symptom intensity but preserved focal examination, and anatomy-driven rationale for definitive clipping).

References	Design/Cohort	Key Population	Therapy	Outcomes	Practice-Relevant Notes
[8]	Rupture-mechanism synthesis	Intracranial aneurysms (incl. ICA–PCom)	—	Conceptual determinants of rupture	Use as background only; avoid patient-level mechanistic claims.
[9]	Imaging + modeling literature	Irregular/multilobulated aneurysms	—	Geometry ↔ heterogeneous flow descriptors	Supports statement that irregular morphology is “higher-risk/less predictable,” but do not imply patient-specific hemodynamics unless performed.
[10]	Clinical–anatomical framework	aSAH with basal cisternal blood	Standard SAH care	Severe symptoms can occur with limited hemorrhage in compact cisterns	Justifies “high symptom intensity despite preserved focal exam” as clinically plausible; keep language clinical, not mechanistic.
[11]	Course/complication context	aSAH across grades	Neurocritical monitoring	Secondary processes (hydrocephalus, sedation, vasospasm/DCI) obscure early exam	Supports value of early bedside phenotype capture in low-grade presentations.
[12]	Endovascular feasibility/limits	Ruptured complex aneurysms (lobulated, daughter sacs, broad neck, branch-adjacent)	Coiling ± adjuncts	Durability/packing challenges increase with complexity	Case-relevant rationale: multilobulation + branch proximity may reduce predictability of complete dome protection.
[13]	Microsurgical durability principle/series	Ruptured aneurysms needing anatomy-driven reconstruction	Microsurgical clipping	Durable exclusion with direct visualization of neck/branches/perforators	Justifies clipping when endovascular durability is less predictable; emphasizes branch/perforator protection.
[14]	ICA–PCom operative corridor literature	ICA–PCom aneurysms (incl. posterior projection/branch-adjacent)	Pterional exposure; carotid/optic windows (as applicable)	Defines safe exposure/dissection logic	Anchors your technical decision-making to established corridors without broad review.
[15]	Complication risk cohorts	aSAH	Standard prevention/monitoring	Rebleeding, vasospasm/DCI, hydrocephalus, CN palsy remain key risks	Frames urgency and why “low-grade” does not mean “low risk.”
[16]	Anatomic risk context	Lesions near AChA/perforators; perforator-rich environments	Endovascular vs. microsurgery	Higher procedural hazard with critical perforators/branching	Supports anatomy-based strategy selection and intraoperative protection priorities.
[17]	Systems/access literature	aSAH across resource settings	Transfer/imaging/ICU pathways	Delays + infrastructure variability worsen outcomes	Supports one concise systems paragraph: time-to-imaging/time-to-securing matters.
[18]	Real-world implementation context	SAH pathways under practical constraints	Flexible protocols	Outcomes depend on matching lesion complexity to expertise/resources	Allows brief acknowledgment of workflow variability without drifting into pharmacology/biomarkers.

## Data Availability

The data presented in this case report are available on request from the corresponding author. The data are not publicly available due to privacy and ethical restrictions, as they contain information that could compromise patient confidentiality.

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
