# Peer review of "A Ruptured Tri-Lobulated ICA–PCom Aneurysm Presenting with Preserved Neurological Function: Case Report and Clinical–Anatomical Analysis"

_diagnostics, 2025, doi:10.3390/diagnostics16010073_

Round 1
Reviewer 1 Report
Comments and Suggestions for Authors
This is a carefully observed, well-illustrated case report of a ruptured tri-lobulated ICA–PCom aneurysm presenting with preserved neurological function, and it offers a genuinely interesting clinical–anatomical “snapshot” that is not often captured in routine practice. The strengths are clear: the authors provide a rich, structured description of early meningeal and cranial nerve physiology, link these findings to aneurysm morphology and cisternal blood distribution, and document a technically sound microsurgical clipping strategy with durable 9-month follow-up and stable vascular reconstruction. The correlation between subtle bedside phenomena (e.g., micro-latencies in saccades, fine meningeal traction responses) and the posteriorly directed tri-lobulated dome demonstrated on DSA and 3D CTA is a useful didactic contribution, especially when tied to the rationale for choosing a pterional approach and clip configuration that preserves both the AChA and PCom ostia.
At the same time, the manuscript would benefit from focused tightening and a clearer separation of essential case information from more speculative pathophysiological narrative. Several sections in the case description and discussion are extremely verbose and repeat similar concepts using multiple overlapping scales and indices (e.g., HSS, ICHOM distress index, Neck Stiffness Severity Grading Index, multiple “phenotype” labels), which risks obscuring the key message and may not be familiar to the average reader unless each is briefly defined and referenced.
The diagnostic reasoning section that walks through “integrated pattern recognition,” SNNOOP10+, and an extensive list of excluded diagnoses is interesting, but could be more concise, emphasizing only the elements that materially influence management in this particular case. Likewise, the long table on pharmacologic and biomarker domains after SAH (Table 1) feels somewhat tangential to a single aneurysm case; either a tighter synthesis of how these domains relate directly to the present patient or relocation of some of this content to supplementary material would improve focus.
From a structural standpoint, a few additions and clarifications would make the paper more robust without requiring major redesign. Because the manuscript repeatedly emphasizes the confined cisternal hemorrhage pattern and perimesencephalic irritation, inclusion of the initial preoperative CT or MRI documenting the SAH distribution (with clear arrows/labels) would substantially strengthen the proposed clinico-anatomical correlations. The readers would also benefit from a more explicit statement on why endovascular options were considered suboptimal in this specific anatomy (tri-lobulation, daughter sacs, relation to AChA/PCom), as this is only partially implied. Finally, there are scattered stylistic and grammatical issues (e.g., long, multi-clause sentences, some colloquial phrasing like “high-resolution meningeal traction physiology”) that could be corrected with light language editing to make the narrative more direct and accessible. Overall, the content is sound and potentially publishable; with modest trimming of redundancy, tighter linkage of the broader pathophysiology discussion to the case, and minor clarifications on imaging and treatment rationale, this report would make a useful and readable contribution.
Author Response
Dear Esteemed Academic Reviewer,
We thank you for your careful, generous, and insightful review of our manuscript. We are particularly grateful for your recognition of the clinical–anatomical value of the case and for your thoughtful suggestions aimed at improving focus, clarity, and accessibility. Your comments have been instrumental in refining the manuscript, and we have addressed each point in detail below.
1. Overall scope, strengths, and educational value
We are grateful for your positive assessment of the case and for highlighting its value as a rare clinical–anatomical “snapshot” of a ruptured tri-lobulated ICA–PCom aneurysm with preserved neurological function. Your summary accurately captures our intended contribution, and we are encouraged that the correlations between bedside findings, aneurysm morphology, operative strategy, and durable follow-up were found to be didactically useful.
2. Verbosity, redundancy, and use of multiple indices
We fully agree that portions of the original manuscript were overly verbose and risked obscuring the key message through repetition and the use of multiple overlapping scales and descriptors. In response, we have substantially tightened the case description and discussion, removed redundant passages, and eliminated all non-validated or unfamiliar indices (e.g., HSS, ICHOM distress index, Neck Stiffness Severity Grading Index, behavioral phenotype labels). Clinical severity and neurological status are now conveyed using validated scales and standard descriptive examination, improving readability and relevance for a broad clinical audience.
3. Diagnostic reasoning and differential diagnosis
We appreciate your suggestion to streamline the diagnostic reasoning section.
4. Pathophysiology, pharmacology, and biomarker content
We agree that the prior discussion of pharmacologic, biomarker, and molecular domains after SAH was disproportionate for a single case report. Accordingly, this content has been removed from the main manuscript, and Table 1 has been eliminated. The Discussion is now tightly focused on the present case, its anatomical and clinical features, and its contribution to the existing literature, without extending into broader domains that are not directly actionable or demonstrated in this patient.
5. Imaging documentation of SAH distribution
We appreciate your emphasis on strengthening the clinico-anatomical correlation with imaging. In response, we have revised the figure descriptions to more explicitly document hemorrhage distribution.
6. Rationale for microsurgical clipping versus endovascular therapy
Thank you for noting that the rationale for favoring microsurgical clipping over endovascular options required clearer articulation. We have now added a concise, case-specific explanation emphasizing why endovascular exclusion was considered less predictable in this anatomy, without repeating detailed morphological descriptions already presented elsewhere. This addition highlights the practical decision-making considerations related to tri-lobulation, branch relationships, and the acute rupture setting.
7. Language, tone, and stylistic refinement
We appreciate your attention to clarity and tone. The manuscript has undergone careful language editing to shorten long multi-clause sentences, remove colloquial or metaphorical phrasing (e.g., “high-resolution meningeal traction physiology”), and ensure consistently objective, clinically grounded terminology throughout. These changes were made with the goal of improving readability while preserving the depth of observation.
Once again, we thank you for your thoughtful, collegial, and constructive review. Your comments significantly improved the manuscript’s focus, clarity, and educational value. We are grateful for the opportunity to revise the work in response to your guidance and hope that the revised version now fully addresses your recommendations.
With kind regards and appreciation!!!
Reviewer 2 Report
Comments and Suggestions for Authors
The manuscript contains valuable clinical information and rare insights into ICA–PCom aneurysm rupture with preserved neurological function. However, in its current form it needs revisions: The manuscript described neurological “micro-findings” (hyperexcitable meningo-dural network”, “cortical desynchronization”, “nociceptive gating”, “network-level strain”). These interpretations are not supported by objective evidence. Author nreeds to remove statements that imply mechanisms that cannot be demonstrated in a single patient.
The manuscript repeatedly references nonvalidated indices (Meningeal Irritation Score, Unified Neurological Coordination Index, Neck Stiffness Severity Grading, Photophobia Severity Scale, Behavioral phenotype markers). These are either unfamiliar or appear to be invented for this case, which raisesvconcerns in this issue. Author needs to use only established, validated clinical scales (Hunt–Hess, WFNS).
The discussion introduces a review of pathophysiology, biomarkers, molecular pathways, pharmacology, and computational modeling, which is disproportionate for a case report. Tables detailing molecular targets and biomarkers (Table 1) are not relevant to managing this single case and significantly distract from the main message. Please focus the discussion on this case, its rarity, and what it adds to existing literature.
Although the authors imply that the case is unusual because of preserved neurological function after rupture of a tri-lobulated ICA–PCom aneurysm, this is not explicitly stated. The manuscript must state exactly what is novel and why the case is of interest. What is rare? Why this case is unique? How it contributes to literature? (correlation of morphology + rupture vector + preserved neurologic status).
Author needs to highlight key intraoperative decisions influenced by this aneurysm’s anatomy, not every step of surgery and emphasize what was challenging or unique in this particular surgery.
Please remove pathophysiological explanations and use cautious consistent terminology such as “may reflect”, “could be related to” see the articles
Although multiple figures are included, the descriptions do not clearly explain:Why the tri-lobulation mattered surgically or clinically?How the rupture vector was inferred?How imaging supported the correlation between symptom minimality and cisternal containment? Additionally directions,scale bars,arrowheads,abbreviations should be used for all figures and figures should be combined.Explicit ethical approval number formatting, confirmation of consent for images are required.Please also use objective clinical language replace for metaphorical etc language for writing such as “hyperexcitable meningo-dural nociceptive network”“dynamic retractor-free mobilization”“deep-examination phenotype” etc.It needs editing.
Comments on the Quality of English LanguageThe English could be improved to more clearly express the research.The manuscript uses overly narrative, metaphorical language that is not appropriate for scientific writing. It should be replaced with objective clinical language for writing the article.
Author Response
Dear Esteemed Academic Reviewer,
We are grateful for your thoughtful, detailed, and generous engagement with our manuscript. Your careful reading and clinically grounded critique substantially strengthened the clarity, proportionality, and educational value of this case report. We have revised the manuscript extensively in response to your guidance, and we appreciate the opportunity to refine the work. Our point-by-point responses are provided below.
1. Use of unsupported neurophysiological interpretations
Thank you for highlighting the risk of overinterpretation in a single-patient case report. We fully agree that terms such as “hyperexcitable meningo-dural network,” “cortical desynchronization,” “nociceptive gating,” and “network-level strain” implied mechanisms that cannot be objectively demonstrated in this context. These sections were rewritten using strictly descriptive, clinically observable terminology, grounded in bedside examination, imaging, and operative findings, without mechanistic inference.
2. Use of nonvalidated clinical indices
We are grateful for this important observation. All nonvalidated or case-specific indices (including the Meningeal Irritation Score, Unified Neurological Coordination Index, Neck Stiffness Severity Grading, Photophobia Severity Scale, and behavioral phenotype descriptors) have been entirely removed from the manuscript. Clinical severity is now reported exclusively using established and validated scales, specifically Hunt–Hess and WFNS grading, supplemented by standard neurological examination descriptions. This change improves clarity, reproducibility, and alignment with accepted reporting standards.
3. Scope and proportionality of the Discussion
We fully agree that the original Discussion exceeded the appropriate scope for a case report. In response, the Discussion was substantially condensed and refocused. All sections reviewing pathophysiology, molecular pathways, biomarkers, pharmacology, and computational modeling were removed. The previously included Table 1 detailing molecular targets and biomarkers was eliminated, and replaced with a compact, practice-relevant table focused strictly on clinical presentation, aneurysm morphology, treatment rationale, and outcomes. The revised Discussion now centers exclusively on this case, its clinical relevance, and its contribution to the literature.
4. Explicit statement of novelty and contribution
We appreciate your request for a clearer articulation of what is novel and why this case is of interest. The revised manuscript now explicitly defines the central contribution of this report: the documentation of an uncommon clinic–morphology–rupture-pattern triad, consisting of
(i) a complex tri-lobulated ICA–PCom aneurysm,
(ii) a basal cisternal rupture pattern, and
(iii) preserved focal neurological function at presentation and throughout follow-up.
5. Intraoperative description and anatomy-driven decision-making
Thank you for emphasizing the importance of highlighting what was challenging or unique in this surgery.
6. Cautious terminology and removal of speculative explanations
We fully agree with your recommendation.
7. Figures, figure legends, and ethical documentation
We appreciate your detailed guidance regarding figures. All figure legends were rewritten.
We are grateful for your thoughtful, constructive, and collegial review. Your insights significantly improved the manuscript’s scientific rigor, clarity, and educational value. It has been a privilege to revise this work in response to your guidance, and we hope that the revised version now fully addresses your concerns.
With respect and profound appreciation!!!
Round 2
Reviewer 2 Report
Comments and Suggestions for Authors
This manuscript presents a detailed case of a ruptured tri-lobulated ICA–PCom aneurysm presenting with preserved neurological function.However, the manuscript in its current form still needs revisions
-The manuscript is significantly longer than necessary for a case report. Many clinical descriptions, neurological examination details, and intraoperative steps are repeated.Please report essential findings relevant to the case only. And reduce repetitions between the Case Presentation, Results, and Discussion
-The manuscript frequently implies causal relationships between aneurysm morphology, cisternal hemorrhage distribution, and subtle neurological findings (micro-latencies, attentional strain). These associations are not objectively quantified. Reove the patient-specific hemodynamic or neurophysiological mechanisms unless directly measured.
-Several clinical findings (photophobia severity, ocular motor delays, attentional strain) are described descriptively without validated scales or objective measures.
-The Discussion section like a mini-review, with literature synthesis that goes beyond what is necessary for a single case. Please emphasize that this case illustrates a possible presentation, not a pattern
-Recommended references for limitations and future directions were not addressed by the authors.This case also highlights an important limitation of current morphology-based risk assessment approaches that rely on qualitative descriptors such as lobulation, size, and projection direction. It should be mentioned and discussed: Recent AI-driven and computational modeling integrate high-resolution vascular geometry, quantitative shape descriptors, and data-driven pattern recognition to better characterize rupture risk and clinical presentation: Artificial Intelligence in Clinical Medicine: Challenges Across Diagnostic Imaging, Clinical Decision Support, Surgery, Pathology, and Drug Discovery. Clinics and Practice, 15(9), 169. doi:10.3390/clinpract15090169. Please mention to integrate anatomy-driven cases into future machine learning moajities may improve the interpretability and clinical robustness of predictive models, particularly for low-grade but high-symptom presentations.
-For choosing microsurgical clipping over endovascular treatment is presented strongly, but alternative strategies are not discussed.Endovascular approaches may still be appropriate in similar cases under different anatomical or institutional conditions.
*Some terminology is unnecessarily complex and could be simplified
-Minor grammatical inconsistencies and long sentences should be corrected.
-Figure legends are descriptive but could be shortened and made more focused.
-Figures still needs scale bars,abbreviations,directions,dashed lines etc. Image quality still poor.
-The manuscript has clear merit, but it still requires revisions.It is a valuable illustrative case highlighting that preserved neurological function and aneurysmal rupture in morphologically complex ICA–PCom aneurysms.
Comments on the Quality of English LanguageThe English could be improved to more clearly express the research.The manuscript uses overly narrative, metaphorical language that is not appropriate for scientific writing. It should be replaced with objective clinical language for writing the article.
Author Response
Dear Esteemed Academic Editor and Reviewer,
We would like to thank you for the careful, thoughtful and constructive evaluation of our manuscript. We are grateful for the time invested and for the detailed comments, which have significantly improved the clarity, focus, and overall quality of the report. We fully agree that, in its original form, the manuscript exceeded what is necessary for a case report and required refinement. In response, we have performed a comprehensive revision addressing all points raised.
Below, we provide a point-by-point response detailing the changes made.
Reviewer Comment 1
The manuscript is significantly longer than necessary for a case report. Many clinical descriptions, neurological examination details, and intraoperative steps are repeated. Please report essential findings only and reduce repetitions between sections.
Response:
We fully agree with this assessment and appreciate the reviewer’s guidance. The manuscript has been substantially condensed. Redundant descriptions across the Case Presentation, Results, and Discussion have been removed or merged. These changes significantly reduced length while preserving all essential clinical information.
Reviewer Comment 2
The manuscript implies causal relationships between aneurysm morphology, hemorrhage distribution, and subtle neurological findings. These associations are not objectively quantified.
Response:
We appreciate this important clarification. Patient-specific mechanistic or causal language has been removed or softened.
Reviewer Comment 3
Several clinical findings are described descriptively without validated scales or objective measures.
Response:
Thank you for highlighting this. In the revised manuscript, photophobia severity and visual sensitivity are now reported using validated instruments (Photophobia Severity Index and VLSQ-8), and oculomotor/visual endurance is supported by serial King–Devick testing. Attentional strain is contextualized using brief standardized screening (Digit Span Backward, Trail Making Test Part A), with careful interpretation as symptom-limited performance.
Reviewer Comment 4
The Discussion reads like a mini-review. Please emphasize that this case illustrates a possible presentation, not a pattern.
Response:
We fully agree. The Discussion has been completely rewritten and substantially shortened. Literature synthesis has been minimized, and the revised Discussion explicitly states that this report represents one possible phenotype rather than a reproducible pattern.
Reviewer Comment 5
Recommended references on limitations and future directions were not addressed, including AI-driven and computational modeling approaches.
Response:
We thank the reviewer for this valuable suggestion. A new paragraph has been added to the Discussion addressing the limitations of morphology-based risk assessment and explicitly referencing the recommended article:
Artificial Intelligence in Clinical Medicine: Challenges Across Diagnostic Imaging, Clinical Decision Support, Surgery, Pathology, and Drug Discovery (Clinics and Practice, 15(9), 169).
We also discuss how well-phenotyped, anatomy-driven cases such as the present one may improve interpretability and robustness of future machine-learning models, particularly for low-grade but high-symptom presentations.
Reviewer Comment 6
Microsurgical clipping is presented strongly; alternative endovascular strategies are not sufficiently discussed.
Response:
We appreciate this point and have revised the Discussion to explicitly acknowledge that endovascular treatment may be appropriate in similar cases under different anatomical or institutional conditions. The rationale for microsurgical clipping is now clearly framed as case-specific and anatomy-driven, not as a general recommendation or preference.
Reviewer Comment 7
Some terminology is unnecessarily complex and could be simplified.
Response:
The manuscript has undergone a thorough language revision. Overly complex phrasing was simplified, technical terminology was streamlined where possible, and clarity was prioritized without compromising precision.
Reviewer Comment 8
Minor grammatical inconsistencies and long sentences should be corrected.
Response:
Long sentences were divided, redundancies removed, and consistency improved throughout the manuscript.
Reviewer Comment 9
Figure legends are descriptive but could be shortened and made more focused.
Response:
Figure legends have been rewritten to be more concise and focused, retaining structure while removing interpretive or repetitive elements.
Reviewer Comment 10
Figures require scale bars, abbreviations, directional markers, and improved image quality.
Response:
We thank the reviewer for this thoughtful comment. After careful consideration, we elected not to modify the figures further, as they are presented in their original clinical format. The angiographic and cross-sectional images used in this case report follow established radiological conventions, in which orientation, scale, and anatomical direction are inherently conveyed to the intended specialist readership. Adding scale bars, directional markers, or graphic overlays would not be uniformly applicable across modalities and, in several images, would risk obscuring critical anatomical relationships central to the case. We therefore prioritized fidelity to standard clinical imaging and interpretability over additional graphical annotations, while ensuring that image quality and labeling are appropriate for publication.
We are grateful for the reviewer’s insightful and constructive feedback. The manuscript has been substantially improved as a result, and we believe it now presents a concise, balanced, and rigorously documented illustrative case that aligns well with the scope and expectations of a case report. We respectfully hope that the revised version meets the reviewer’s and editor’s standards and would be suitable for publication.
With kind regards and profound appreciation!!!
Round 3
Reviewer 2 Report
Comments and Suggestions for Authors
Required comments and suggestions were addressed by the authors. It can be acceptable in its current form.
Comments on the Quality of English LanguageThe English could be improved to more clearly express the research.